



# Pliocene shorelines and the epeirogenic motion of continental margins: A target dataset for dynamic topography models

Andrew Hollyday[1], Maureen E. Raymo[1], Jacqueline Austermann[1], Fred Richards[2], Mark Hoggard[3], Alessio Rovere[4]

[1]Department of Earth & Environmental Sciences, Columbia University & Lamont-Doherty Earth Observatory, New York, USA
[2]Department of Earth Science & Engineering, Imperial College London, London, UK
[3]Research School of Earth Sciences, Australian National University, Canberra, AU
[4]DAIS Department for Environmental Sciences, Informatics and Statistics, Ca' Foscari University of Venice, Venice, IT

*Correspondence to*: Andrew Hollyday (andrewh@ldeo.columbia.edu)

**Abstract.** Global mean sea level during the mid-Pliocene Epoch (~3 Ma), when $CO_2$ and temperatures were above present levels, was notably higher than today due to reduced global ice sheet coverage. Nevertheless, the extent to which ice sheets responded to Pliocene warmth remains in question, owing to high levels of uncertainty in proxy-based sea-level reconstructions as well as solid Earth dynamic models that have been used to evaluate a limited number of data constraints. Here, we present a global dataset of ten wavecut scarps that formed by successive Pliocene sea-level oscillations and which are observed today at elevations ranging from ~6 to 109 m above sea level. The present-day elevations of these features have been identified using a combination of high-resolution digital elevation models and field mapping. Using the MATLAB interface TerraceM, we extrapolate the cliff and platform surfaces to determine the elevation of the scarp toe, which in most settings is buried under meters of talus. We correct the scarp-toe elevations for glacial isostatic adjustment and find that this process alone cannot explain observed differences in Pliocene paleoshoreline elevations around the globe. We next determine the signal associated with mantle dynamic topography by back-advecting the present-day three-dimensional buoyancy structure of the mantle and calculating the difference in radial surface stresses over the last 3 Myr using the convection code ASPECT. We include a wide range of present-day mantle structures (buoyancy and viscosity) constrained by seismic tomography models, geodynamic observations, and rock mechanics laboratory experiments. Finally, we identify preferred dynamic topography change predictions based on their agreement with scarp elevations and use our most confident result to estimate a Pliocene global mean sea level based on one scarp from De Hoop, South Africa. This inference (11.6 ± 5.2 m) is a downward revision and may imply ice sheets were relatively resistant to warm Pliocene climate conditions. We also conclude, however, that more targeted model development is needed to more reliably infer mid-Pliocene global mean sea level based on all scarps mapped in this study.

## 1 Introduction



While projections of GMSL rise by 2050 show relatively good agreement with one another, they diverge significantly by 2100, spanning a range of <0.5 m to >2.5 m (DeConto et al., 2021; Masson-Delmotte et al., 2021). This wide range reflects greater sensitivity of GMSL to factors such as the emission scenario and the mechanical processes that control ice-sheet stability
(Masson-Delmotte et al., 2021). To reduce uncertainty in sea-level projections for 2100 and beyond, some studies have used ice-sheet models forced under paleoclimate conditions to identify which model parameters can reproduce melt responses consistent with sea-level constraints from the geologic past (DeConto et al., 2021; Edwards et al., 2019; Masson-Delmotte et al., 2021). As such, the confidence of near-future sea-level projections depends on the level of confidence and uncertainty associated with sea-level change in the past.

The Pliocene Epoch (5.3 to 2.6 Ma), which consisted of a climate similar to present and projected conditions, may serve as a testing ground to better understand Earth's imminent climate future and has been widely used to calibrate dynamic ice-sheet models (Burke et al., 2018; DeConto et al., 2021; Dumitru et al., 2019). The Mid Pliocene Warm Period (MPWP; ~3.3 to 2.9 Ma), in particular, was characterized by maximum warm period temperatures 2.5°C to 4°C above the 1850 to 1900 baseline and had $CO_2$ concentrations close to modern values (~360 to ~420 ppm; Fedorov et al., 2013; Fischer et al., 2018;
Haywood et al., 2013). Polar amplification was substantial as well with temperatures ~8°C above the 1850 to 1900 baseline in high-latitude regions (Fischer et al., 2018). While quantifying the melt response of ice sheets under these warm conditions can provide important model parameterization, to date, estimates of GMSL during the MPWP relative to present span a substantial range of ~5 to 25 m (Masson-Delmotte et al., 2021). For example, Dumitru et al. (2019) constrained GMSL between 5.6 m and 19.2 m (16[th] and 84[th] percentiles) using phreatic overgrowth on speleothems from Mallorca, Spain, which date to 3.27 ±
0.12 Ma and were corrected for solid Earth deformation since their formation. That study is currently the only inference for the MPWP that accounts for various Earth deformation processes and rigorously propagates and reports uncertainty; however, GMSL estimates from coastal features from South Africa (~14 m; Hearty et al., 2020), the United States (~15 m; Moucha & Ruetenik, 2017), and New Zealand (~25 m; Grant et al., 2019) generally support this range.

         Constraining GMSL millions of years back in time poses several distinctive challenges. While oxygen isotope-based
sea-level reconstructions offer the advantage of being continuous through time, they consist of inherently high uncertainties (~10 to 13 m) due to ocean temperature effects, diagenetic processes, and unconstrained sea water chemistry (Grant et al., 2014; Raymo et al., 2018; Rovere et al., 2014; Shakun et al., 2015; Spratt & Lisiecki, 2016). Alternatively, relic coastal features (i.e., paleoshorelines) along the margins of the global oceans outline past sea-level changes. By correcting for post-depositional effects such as sediment compaction, crustal deformation, glacial isostatic adjustment (GIA), and mantle dynamic topography
(DT; i.e., surface deflections caused by mantle convection), one can constrain a GMSL offset from present day.

         Models of GIA, a process that describes the viscoelastic response of the solid Earth, its gravity field, and rotation axis, to changes in ice and ocean loads (Farrell & Clark, 1976), consist of uncertainties due to the ice-loading history and the mantle rheological structure. GIA models produce up to meters-scale uncertainties that are greatest in the near field and smallest in the far field of ice-sheet loading. Simulations and observations of DT change suggest convection can cause surface
deflections at rates as high as >100 mMyr[-1] (Austermann et al., 2015; Czarnota et al., 2013; Guiraud et al., 2010; Hoggard et



al., 2016; Hollyday et al., 2023; Roberts & White, 2010). As a result, ancient shorelines have undergone significant spatiotemporally variable deformation due to DT change, even at passive margins, which were once thought to be relatively stable (Austermann et al., 2017; Moucha et al., 2008). However, uncertainties in these models are on the order of 10s to 100s of meters due the presence of many more free and unconstrained parameters associated with the mantle's rheological, viscosity,

density, compositional, and temperature structure. Nonetheless, the epeirogenic motion of paleoshorelines can be corrected for using geodynamic predictions, revealing information about past ice-sheet and sea-level change from direct geologic evidence (e.g., Hollyday et al., 2023; Moucha & Ruetenik, 2017).

        Rovere et al. (2014) reported three wavecut escarpments from Australia, the United States, and the Republic of South Africa, which were all carved during Pliocene times by the continuous, low amplitude oscillations in sea level. These

oscillations are inferred from the benthic $\delta^{18}O$ record, a climate proxy that shows ice volume and temperature always returned to approximately the same level over more than two million years of orbitally-paced climate cycles from approximately 5 Ma to 3 Ma (Lisiecki & Raymo, 2005). After ~2.9 Ma, global ice volume expanded permanently due to the intensification of northern hemisphere glaciation, and GMSL would have retreated seaward, stranding these former sea cliffs inland. Late Pliocene intertidal to subtidal facies are found at the base of the scarps (Dowsett & Cronin, 1990; James et al., 2006; Rovere

et al., 2014). This scarp formational process is consistent with simulations of marine terrace formation by Trenhaile (2014), particularly in settings where relatively slow uplift has occurred.

        Here, we augment the initial Rovere et al. (2014) database consisting of three Pliocene scarps by including seven additional sites found using high-resolution topography data. These sites occur on continental margins characterized by a flat coastal plain bounded by a steeper scarp. In each case, the coastal plains extend for tens to hundreds of kilometers inland from

the ocean and are mantled with late Pliocene to Pleistocene marine sediments that extend to the scarp "toe," or the intersection between the scarp cliff and platform surfaces. Older, typically Miocene facies occur inland from the scarp toe (Fig. 1). These roughly shore-parallel scarps, which range from a few 10s of km to >1000 km in length, have all undergone some combination of erosion, uplift, subsidence, and isostatic adjustment since their formation. Here, we compare the observed elevations of these paleoshoreline features to predictions of GIA and DT (Raymo et al., 2011; Hollyday et al., 2023). The total vertical and

along-scarp displacement of a shoreline from a horizontal, eustatically-controlled baseline provides constraint that allows us to evaluate DT model output, information that ultimately will lead to the improvement of such models as well as the eventual isolation of a common Pliocene GMSL signal from a global observational database.






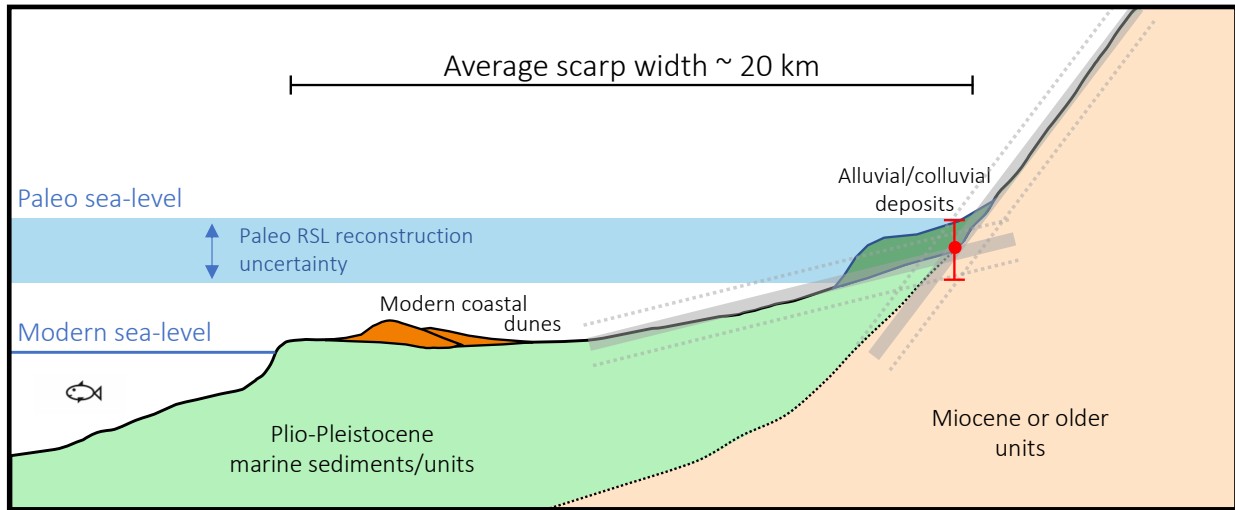

**Figure 1: Schematic illustration of coast-perpendicular scarp profile. The grey lines (dashed lines indicate 2σ**
**uncertainty bands) represent the TerraceM analysis, where the cliff and platform surfaces have been extrapolated to**
**determine their geometric intersection, which often is buried under meters of alluvial/colluvial material (colored in**
**dark green). The red point corresponds to the scarp toe elevation, and the red uncertainty bar describes the RSL**
**estimate before the indicative range, GIA, or DT change corrections have been applied.**

## 2 Methods

### 2.1 Initial search for Pliocene scarps

To guide a global search for Pliocene-aged scarps, we used the EarthEnv-DEM90 digital elevation model (DEM) dataset,
which has a lateral resolution of 90 m (Robinson et al., 2014). Using the topographic modeling tool in the ENVI software
package, we built a 1 arc-degree pixel-scale raster that represents land-surface slope on continental margins (<100 km from
the sea). This tool compares elevation values across a 3 x 3-pixel grid. Note that since the surface slope is calculated from this
grid, geomorphological features that have a lateral extent less than the pixel size of ~270-300 m are not resolved. From this
slope map, we identified coastal plains that had low slopes indicative of long-term coastal erosion down to a marine platform.
We chose a slope value of 0.625 (mkm$^{-1}$) since this is lower than the average slope of the coastal plains located seaward of the
three Pliocene scarps mapped in Rovere et al. (2014) and is also consistent with continental shelf clinoform angles as modeled
by Pirmez et al. (1998). We then eliminated all locations where the general topography did not exhibit a relatively steep inland
scarp with a high rim bounding the coastal plain. Additionally, any such slope breaks (e.g., scarp) that was less than 20 km in
length was excluded in order to focus on longer wavelength topographic features, hypothesized to be paleoshorelines that can
be used to assess long-term deformation of continental margins. We consider the toe of each scarp to represent a former sea-
level highstand (Fig. 1). Our analysis includes only scarp/plain regions where independent literature and/or geologic maps
confirm that shallow marine sediments of Plio-Pleistocene age blanket the coastal plain. Finally, we classify the quality of
each scarp's geochronology with qualitative descriptors: weak, moderate, or strong (Fig. 2). A weak constraint lacks absolute



and robust relative age control; a moderate constraint consists of a robust relative geochronology; and a strong constraint requires absolute age control. Classifying the formational age uncertainty of the scarps ultimately informs our data-model comparison as our geodynamic models do not explicitly account for this source of uncertainty. All age control qualifiers reflect an estimate of the confidence of the geochronology relative to mid-Pliocene times (3 Ma).

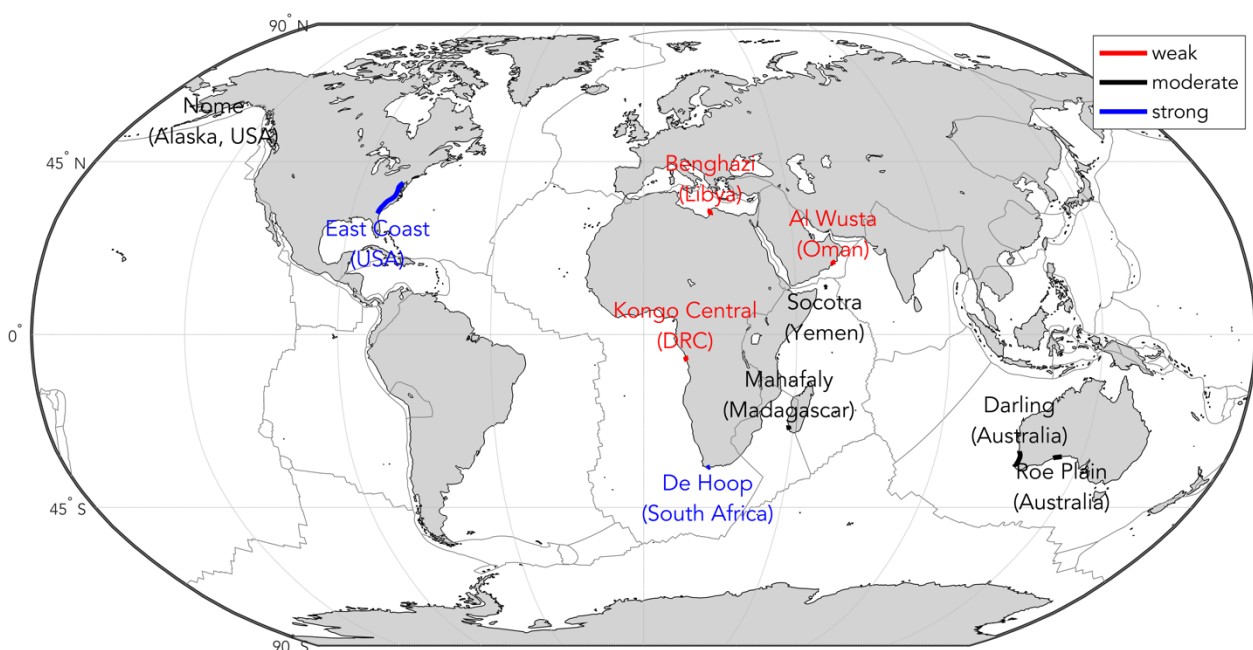


**Figure 2: Global compilation of ten Pliocene-aged scarps. Colors indicate the quality of the age constraint relative to the MPWP (3 Ma).**

### 2.2 High-resolution scarp mapping

Following the initial global search for Pliocene-aged scarps, we used higher resolution DEMs from the Shuttle Radar

Topography Mission (SRTM) which is resolved at 1 arc-second (30 m; Farr et al., 2007). Vertical errors were assessed from ground truthing and are <10 m globally, with the greatest uncertainties in places with high relief (Rodríguez et al., 2006, 2005). We note that the Nome (Alaska, USA) site was not analyzed in high resolution as SRTM data is not available at that latitude. Points were approximately chosen every 30 m (resolution of DEMs) along each scarp where a clear break in slope occurs. These points represent the approximate intersection of scarp cliff and platform surfaces, or the scarp toe (Fig. 1). Next, we

defined profiles orthogonal to the scarp that pass through each point along the scarp to characterize the geomorphological structure. Using the previously defined points, we visually selected the base of the escarpment cliffs and extracted the elevation. While this secondary procedure places a more precise constraint on remotely sensed elevations compared to the initial global search, ongoing alteration of the surface geomorphology owing to stream incision, flexural and brittle crustal deformation, and



colluvial/alluvial deposition are readily apparent in almost all the DEMs that we evaluate (Fig. 1 & 3). Given these factors, we
assigned points away from regions clearly affected by stream incision and/or faulting, though these processes can be
challenging to uniquely identify from DEMs. To constrain the extent to which colluvium has buried the scarp toes, we
performed an additional analysis, which is described in Section 2.3. We did not quantitively consider the role that sedimentary
loading changes may have had on isostatic and flexural deformation of the scarps; however, the GIA correction (Section 2.4.1)
that we applied includes flexural deformation due to sea-level (or ice load) changes since the time of each scarp's initial
formation.



**Figure 3: Compilation of DEMs for Pliocene-aged scarp database. (a-h): DEMs from the SRTM (Farr et al., 2007). (i & j): DEMs from the General Bathymetric Chart of the Oceans (GEBCO) 2023 grid (GEBCO Compilation Group, 2023). GEBCO DEMs were not used in our analysis of scarp geomorphology. Black circles indicate distance zero of the along-scarp profile, and the yellow dashed lines trace the along-scarp profile. Teal lines show the specific locations of the profiles perpendicular to the scarp, which were analyzed with TerraceM and are shown in Figure 4.**

## 2.3 TerraceM mapping and indicative meaning

To determine the intersection between the scarps' cliff and platform surfaces, we employed the MATLAB interface TerraceM, which is a tool developed to quantitatively assess how various surface processes have affected topographic



evolution (Jara-Muñoz et al., 2016). We used TerraceM to determine the elevation of the scarp toe for each cross section defined orthogonal to the scarp. TerraceM identified maximum elevation profiles perpendicular to the scarp within a swath that was drawn over the previously assigned along-scarp points. The maximum elevation profile was used because this reflects the surface that has been eroded the least and likely corresponds to the most accurate depiction of paleo-sea level. Next, using TerraceM, we computed linear regressions for each cliff and platform surface and extrapolated them to find their intersection, or the scarp toe, buried beneath alluvium or colluvium (Fig. 1). Vertical uncertainties of the intersection were computed from extrapolated 2σ ranges in the linear regressions (Jara-Muñoz et al., 2016). To note, only a small subset of elevation profiles showed paradigmatic geomorphology, with clearly defined cliff and platform surfaces. In most cases, the structure was more complex, reflective of multiple surface and neotectonic processes acting together. These include aeolian deposition (i.e., dunes), tectonic faulting, fluvial incision, and alluvial and colluvial deposition, and were important factors taken into consideration while defining the cliff and platform surfaces. Given the presence of surface deposition along many of the scarps, in almost all cases our TerraceM analysis adjusted elevations downward.

The TerraceM analysis allowed us to calculate at each point along the scarp profiles the elevation of the inner margin of the marine terrace (in some geographic areas referred to as the "shoreline angle," see Davis (1933) and Muhs (2022)), which is commonly used as a proxy for former relative sea level. However, the relationship between the elevations of the inner margin (i.e., scarp toe) and the former sea-level position needs to be quantified through the calculation of the indicative meaning (Kelsey, 2015), which is defined by the reference water level and the indicative range (Rovere et al., 2016; Shennan, 2015; van de Plassche, 2013). These two elements define the position of the paleo-relative sea level with respect to the measured sea-level index point and associated uncertainty. In the absence of modern analogues (i.e., direct measurements of the inner margin of modern coastal sites within the areas of interest), we adopted the values calculated by the software IMCalc (Lorscheid & Rovere, 2019) in each area. IMCalc calculates the reference water level and indicative range for a marine terrace at a given location by extracting local wave and tidal data. From these inputs, the highest reach of storm waves (storm wave swash height) and the breaking depth of ordinary waves are calculated and used as, respectively, upper and lower limits of the indicative range.

## 2.4 Geodynamic models

### 2.4.1 Glacial isostatic adjustment

Each scarp was corrected for GIA following the approach of Raymo et al. (2011), which predicted disequilibrium due to both ongoing solid Earth, gravitational, and rotational adjustment from ice sheet variations since Marine Isotope Stage (MIS) 5e (~122 ka) to present day as well as ice-sheet variations from the Pliocene Epoch (~3 Ma). Here, we only account for the former, since they produce the larger signal and do not depend on assumptions about mid-Pliocene melt geometries. The time-evolving ice sheet geometries for the most recent disequilibrium (122 ka to present day) are from the ICE-5G model (Peltier, 2004), which was extended back to MIS 5e (see Raymo et al., 2011). While newer ice histories than ICE-5G have been published,





most of the scarps are insensitive to the details of the ice history. The exceptions are scarps close to former ice sheets, for
which a more detailed GIA analysis should be considered in future work. This correction uses 36 radial Earth structures that
consist of two lithospheric thicknesses (71 and 96 km), three ($0.1 \times 10^{21}$, $0.3 \times 10^{21}$, $0.5 \times 10^{21}$ Pa s) upper mantle viscosities,
and six ($3 \times 10^{21}$, $5 \times 10^{21}$, $7 \times 10^{21}$, $1 \times 10^{22}$, $2 \times 10^{22}$, and $3 \times 10^{22}$ Pa s) lower mantle viscosities. While radial Earth structures
neglect known lateral variations in mantle viscosity (e.g., Ritsema et al., 2011), we tested an ensemble of possible mantle
structures to estimate the uncertainty ($1\sigma$) associated with an assumed 1-D structure. The final GIA correction is the mean and
standard deviation of the ensemble.

### 2.4.2 Dynamic topography change

After correcting the elevations of each scarp for GIA, we compared the remaining deformation to a suite of DT change
predictions based on models from Hollyday et al. (2023) and additional models parameterized with the TX2008 mantle
structure (Simmons et al., 2009). To compute DT change over the last 3 Myr, we solved the governing equations for mantle
convection, conservation of mass, energy, and momentum, and back-advected the present-day buoyancy structure of the mantle
using the finite element convection code ASPECT (Bangerth et al., 2020; Heister et al., 2017; Kronbichler et al., 2012). Using
ASPECT 2.2.0, we ran global incompressible simulations (using the Boussinesq approximation) with surface and core-mantle-
boundary (CMB) thermal boundary conditions set to 0°C and 3027°C, respectively. We set values for reference temperature
(1333°C), reference density (3300 kg m$^{-3}$), and specific heat (1250 J K$^{-1}$ kg$^{-1}$). Depth-dependent gravity and thermal expansivity
profiles were adopted from Glišović & Forte (2015), and thermal diffusivity was set to zero, since diffusion is not a time-
reversible process and is expected to be minimal over the timespan of 3 Myr. We did not include deflection of internal
boundaries, thermal boundary layers, internal radiogenic heat production, or phase changes within our models. We paired three
different radial viscosity profiles with nine different 3-D buoyancy structures to produce 27 total time-dependent simulations
and accounted for lateral viscosity variations within our simulations through an Arrhenius relationship. Additional details of
the model setup and parameters can be found in Hollyday et al. (2023).

The 3-D structure of the mantle is becoming increasingly well-resolved through seismic tomography (e.g., Lei et al.,
2020; Schaeffer & Lebedev, 2013); however, subtle differences in the amplitude and extents of key structures occur across
models. The initial tomography-derived buoyancy structure exerts a first-order influence on global convection styles and
patterns of DT (Flament et al., 2013). To probe this uncertainty, we computed nine temperature models from shear-wave
velocities. In the upper mantle, we computed temperatures from two high-resolution tomography models: GLAD-M25 and
SL2013sv (Lei et al., 2020; Schaeffer & Lebedev, 2013). Temperatures were computed using an experimentally and
observationally constrained conversion that accounts for the anelastic behavior of the upper mantle (Richards et al., 2020;
Yamauchi & Takei, 2016). In the transition zone and lower mantle (below 410 km), we computed temperatures from four
global tomography models: TX2011, GLAD-M25, S362ANI+M, and S40RTS (Grand, 2002; Lei et al., 2020; Moulik &
Ekström, 2014; Ritsema et al., 2011). Using the linear, depth-dependent conversion factor from Steinberger (2016), we



converted relative seismic velocities to relative density variations. We combined each upper and lower mantle temperature structure to yield eight temperature fields. Additionally, we used the density field, TX2008, which is a joint inversion of geodynamic and seismic data (Simmons et al., 2009). The full conversion scheme for the construction of our initial temperature models, excluding those computed from the TX2008 density model (Simmons et al., 2009), is available in Supplementary

Figure 1 of Hollyday et al. (2023).

We computed DT from the radial surface stresses that result within our mantle convection simulations and the density contrast between the crust and overlying material (water or air; Zhong et al., 1993). To account for overburden changes from air to water along the coasts and over the ocean basins, we applied the formalism of Austermann & Mitrovica (2015) using a 90 km-thick lithosphere. We also computed perturbations to the Earth's geoid, as we are interested in sea-level change instead

of just solid Earth deformation (geoid changes are included in "DT changes" for the remainder of the article). Displacement of DT fields over 3 Myr was accounted for with a plate motion correction, which is described in Hollyday et al. (2023). After performing the plate motion correction for each simulation, the total suite consists of 135 predictions of DT change.

To compare the data to model predictions, we first corrected observations for GIA and propagated uncertainties from GIA, elevation uncertainty, and indicative range. We then corrected each elevation for DT (one DT model at a time) and

calculated a weighted mean GMSL and weighted standard deviation of GMSL for each site, where weights scale with the inverse of the uncertainty of each datapoint. If the uncertainty is small, it indicates that GIA- and DT-corrected scarp elevations form a mostly flat surface, which indicates that the DT prediction is more likely. Another way to quantify this aspect is to calculate the Mean Weighted Standard Deviation (MSWD) of the GIA- and DT-corrected scarp elevations: the smaller the MSWD, the smaller the variability of GIA- and DT-corrected elevations and the higher the confidence in a specific DT model.

Details on the procedure for correcting scarps for GIA, propagating uncertainties, calculating MSWD, and computing an inferred GMSL value can be found in Hollyday et al. (2023). We segregated DT change models into two groups, based on (1) weak MSWD (<30) and GMSL (0 m to 50 m) and (2) stringent MSWD (<20) and GMSL (10 m to 40 m) thresholds. The stringent thresholds were chosen with the goal of identifying models that have a high level of agreement with the observations and fall within existing constraints of Pliocene GMSL.

**3 Results**

We identified seven linear geomorphic features where an abrupt break in slope marks the transition between Plio-Pleistocene and older, inland rocks. These seven scarps are in addition to the three described in Rovere et al. (2014) and Rovere et al. (2015) to a make a total of ten in the dataset. See Supplementary Figure 1 for reported scarp elevations, indicative range, GIA correction, and the data source specific to each location. Each of the ten scarps described here exhibits short to long-wavelength

patterns of solid Earth deformation that have evolved over millions of years. We have detailed this observed deformation in the context of each respective local geologic and geomorphic setting. Next, we have corrected the observed deformation of the





scarps for GIA and compared the remaining deformation to a subset of preferred DT change models (those that pass the weak and stringent thresholds).

## 3.1 Scarp geomorphology

### 255     3.1.1 Darling (Australia)

The Darling (Australia) scarp is located in Western Australia (Fig. 2). The coastal plain is covered by Quaternary sediments of the Kwinana Group. The Bassendean Sands occur within this group and have been attributed to estuarine, shallow marine, and fluvial depositional environments. At the base of the scarp, the Yoganup Formation is characterized as a shoreline facies (Kendrick et al., 1991). In the northeast, the north-south trending Darling Fault divides these sedimentary units from the

Darling plateau, where Archean to Cenozoic metamorphic and intrusive bodies occupy the higher elevation terrain (Raymond et al., 2012). The scarp coincides with the Darling Fault in its central part but diverges from it to the north and south (Fig. 3a). Several alluvial fans cover the scarp's toe, and four major rivers incise the paleoshoreline (Moore, Swan, Canning, and Brunswick Rivers). Absolute age control has not been established for this scarp, but the paleoshoreline is straddled by rock units that predate and postdate Pliocene times (Kendrick et al., 1991). As such, we classify its age control as moderate (Fig.

2). Remotely sensed elevations mapped in high resolution and analyzed with TerraceM (Fig. 4a) show short-wavelength patterns of deformation varying from 30 m to 105 m above present sea level along the length of the roughly north-south trending scarp. From north to south, elevations increase until ~70 km distance, before decreasing to the minimum mapped elevation at ~300 km distance; from ~300 km to the southern terminus of the scarp (469 km distance), elevations increase modestly (~15 to 20 m; Fig. 5a).




**Figure 4: Scarp-perpendicular elevation profiles analyzed with TerraceM (Jara-Muñoz et al., 2016). Locations of each scarp are shown as teal profiles in Figure 3a-h. The black solid and dashed lines show the extrapolated platform and cliff surfaces and their 2σ uncertainties, respectively. The blue points correspond to the extrapolated intersection of**
**the two surfaces.**



**Figure 5: Along-scarp elevations profiles without geodynamic corrections. Red points indicate the mapped elevations of each scarp with uncertainties associated with the TerraceM analysis (Jara-Muñoz et al., 2016), field measurements, and indicative range. Gray line shows Gaussian Process Regression model trained by the data. Dark and light gray bands correspond to 1σ and 2σ uncertainties of the model, respectively. (a) Darling (Australia); (b) Roe Plain (Australia); (c) Kongo Central (DRC); (d) Benghazi (Libya); (e) Mahafaly (Madagascar); (f) Al Wusta (Oman); (g) De Hoop (South Africa); (h) Socotra (Yemen); (i) Nome (Alaska, USA); (j) East Coast (USA); (k) map inset showing site locations.**



### 3.1.2 Roe Plain (Australia)

The Roe Plain (Australia) scarp, also known as the Hampton Escarpment, is located in Western Australia near the Great Australian Bight (Fig.3). The scarp's bedrock geological and geomorphological structure, which consists of Cenozoic limestone at the cliff and a Plio-Pleistocene shallow marine assemblage at the platform (James & Bone, 2007), is described in detail in Rovere et al. (2014). We characterize this scarp, which has been observed in the field (Rovere et al., 2014), with moderate age control due to the availability of a robust relative geochronology (Fig. 2). The Roe Plain (Australia) scarp is mapped with a combination of high-resolution remote sensing, TerraceM analysis (Fig. 4b), and direct field measurements. It extends 260 km in length and varies from 9.5 m to 32.8 m in elevation. The western beginning of the scarp trace marks its lowest point; from there elevations increase until ~75 km distance, where the maximum elevation occurs. From ~75 km to ~260 km distance, the elevations gradually decrease by ~20 m (Fig. 5b). Colluvial deposition at the scarp's toe is corrected for with our TerraceM analysis with corrections typically <10 m (Fig. 4b).

### 3.1.3 Kongo Central (DRC)

The Kongo Central scarp occurs along the Atlantic coast on the border of the Democratic Republic of the Congo (DRC) and Angola and is dissected by the Congo River, which is the second largest river in the world and has an average discharge of 45,000 $m^3 s^{-1}$ (Fig. 3c; Eisma & Van Bennekom, 1978). Nairn & Stehli (1982) describe the scarp's geology as Precambrian and Cretaceous in age, covered by Plio-Pleistocene sedimentary units. They report two to three marine terraces assumed to have formed over Pleistocene sea-level oscillations, though the structure of these terraces remains enigmatic from DEM mapping (Fig. 3c). As only a relative geochronology is available for this site, the age constraint of this scarp is characterized as weak (Fig. 2). The elevations of the Kongo Central (DRC) scarp, which are mapped in high resolution and analyzed with TerraceM, show a mostly linear increase along the length of the scarp (155.6 km; Fig. 5c) with minimum and maximum elevations estimates of 12.7 m to 71.8 m, respectively.

### 3.1.4 Benghazi (Libya)

The Benghazi (Libya) coastal plain is located along the northeastern coast of Libya, the eastern coast of the Gulf of Sirte, and the western margin of the Cyrenaica platform and the Jabal Al Akhdar uplift area (Fig. 3d; Fiduk, 2009). The capital city of Benghazi is situated on this coastal plain approximately 25 km to the west of the scarp. The Cyrenaica platform and Jabal Al Ahkdar bedrock, which compose the high elevation cliff terrain of the Benghazi (Libya) scarp are of Miocene and upper Cretaceous age, respectively (Hallett, 2002). The scarp is interrupted by a large colluvial deposit in its central part (Figs. 3d & 4d). The coastal plain consists of Pliocene sandstone with ichnofacies suggestive of a lower intertidal to shallow subtidal depositional environment (Kumar, 2015). We note that while these coastal plain sediments have been interpreted to be of Pliocene age, uncertain stratigraphic correlations for Quaternary and Pliocene units (Tawadros, 2001, 2011) as well as the



absence of absolute age constraints have led to age control that can generally be characterized as weak (Fig. 2). The elevations
of the Benghazi (Libya) scarp are mapped in high resolution and analyzed with TerraceM (Fig. 4d). The full length of the scarp
is ~191 km, beginning close to the coast (~8 km) in the north and extending inland as far as ~50 km in the south. The scarp is
characterized by considerable (~50 m, vertically) short-wavelength (10s of km, spatially) variability (Fig. 5d). The
northernmost mapped elevations are as low as 45.6 m, and to the south the scarp's elevations increase to its maximum elevation
at ~130 km distance (109.1 m elevation) before decreasing modestly until the terminus of the scarp trace in the south. Given
the presence of well-developed colluvial and/or alluvial structures along the scarp's toe, the TerraceM analysis, which is well
suited for this site, adjusts the scarp's elevations downwards by ~20 to 30 m (Fig. 4d).

### 3.1.5 Mahafaly (Madagascar)

The Mahafaly (Madagascar) scarp is located in the southwest of Madagascar and borders the Mahafaly plateau to the east and
a broad coastal plain to the west (Fig. 3e). Lake Tsimanampetsotsa occurs adjacent to the scarp ~55 km distance from the
north. According to state geological maps, the surface sediments on this coastal plain are mostly composed of unconsolidated
sands of Quaternary age (Dandouau, 1922; Nairn & Stehli, 1982; Persits et al., 1997). Field mapping in June 2016 confirms
that along the coastal plain, the bedrock consists of sub-horizontal shallow-marine, shell-rich limestone (Fig. 6). This scarp is
marked by a lithological contact between the platform and cliff rocks with the cliff consisting of Tertiary limestones, marls,
and chalks (Moat & Du Puy, 2010). Several dune fields occur across the coastal plain and adjacent to the scarp's toe in the
south. While we lack an absolute geochronology for this site, the approximate relative chronology provides a moderate
constraint on the scarp's age (Fig. 2). The elevations of the Mahafaly (Madagascar) scarp are mapped in high resolution and
analyzed with TerraceM (Fig. 4e). In total the scarp extends ~129 km and ranges in elevation from 5.5 m to 31.5 m, with its
minimum and maximum elevations occurring in the scarp's center (~56 km distance) and southern terminus (~129 km
distance), respectively. The mapped deformation is characterized by a slight downward dip in elevation at its center with higher
elevations to the north and south (Fig. 5e).



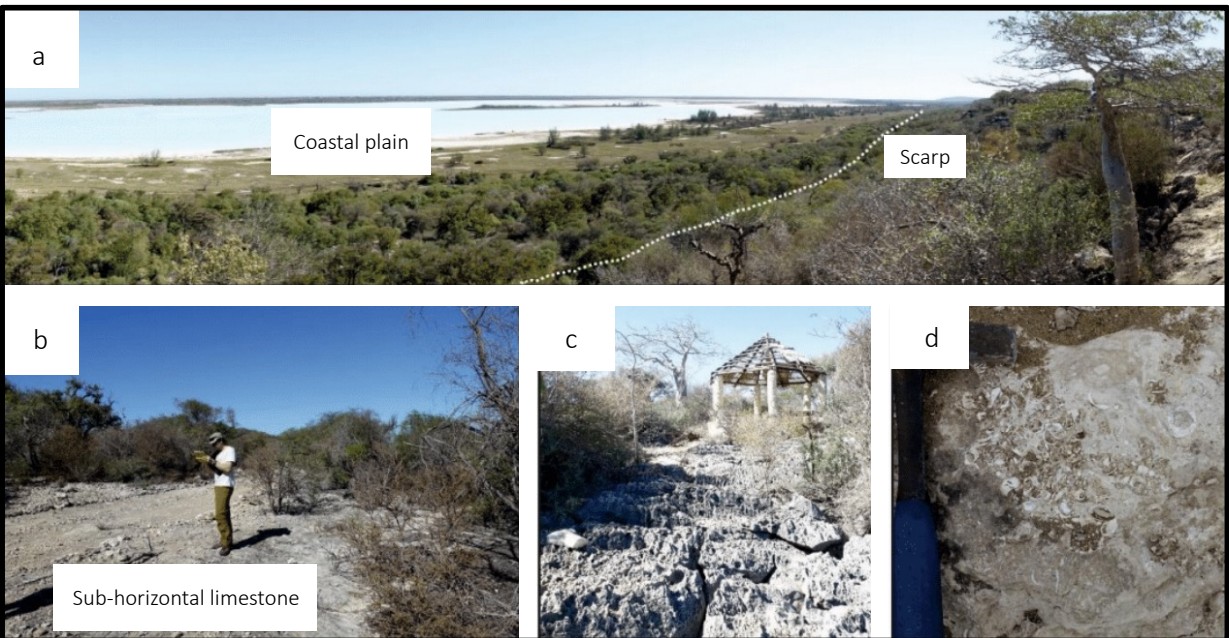

**Figure 6: Field photos from Mahafaly (Madagascar) scarp taken in 2016. (a): View from the top of the cliff, looking seawards. (b): Sub-horizontal limestone at the base of the scarp. (c): Limestone at the top of the Mahafaly cliff. (d): Detail of the shell-rich limestone outcrop on the coastal plain, between the inner margin of the scarp and the coast.**

### 3.1.6 Al Wusta (Oman)

The Al Wusta (Oman) scarp is located in the Al Wusta governorate along the southern coast of Oman (Fig. 3f). This site lies to the south of the Hugf-Haushi Uplift and to the east of the South Oman Salt Basin (Grosjean et al., 2009; Robertson et al., 1990). Reverse faults and synclinal structures, that formed during Tertiary times, occur to the north and south of the scarp (Abbasi et al., 2013; Fournier et al., 2004; Hanna, 1990; Ries & Shackleton, 1990). The coastal plain sediments are classified as Quaternary in age, and the bedrock that composes the scarp cliff and elevated plateau are of Miocene or older origin (Fournier et al., 2004; Nairn & Stehli, 1982; Platel et al., 1994). Two large alluvial fans disrupt the trace of the scarp toe at the Wadi Ainina and Wadi Watif. With the lack of an absolute or robust relative age chronology, we classify this scarp as weak (Fig. 2). The Al Wusta (Oman) scarp has been mapped in high resolution and analyzed with TerraceM (Fig. 4f). The full length of the scarp extends ~138 km and varies in elevation from 13.7 m to 32.6 m. Given the presence of large alluvial and colluvial structures across the scarp, our TerraceM analysis provides significant improvement to our elevation estimates with downward corrections on the order of 10s of meters (Fig. 4f). This scarp exhibits a slight decrease in elevation from ~30 m in the south to ~20 m in the north, which mostly occurs between distance 40 km and 80 km (Fig. 5f).



### 3.1.7 De Hoop (South Africa)

The De Hoop (South Africa) scarp is located on the southern coast of the Republic of South Africa, ~160 km southeast of Cape Town (Fig. 3g). The coastal plain consists of the De Hoopvlei Formation (shallow water depositional environment; Malan, 1991). An oyster shell within the unit dates (Strontium Isotope Stratigraphy; SIS) to 3.56 ± 1.08 Ma and confirms this biostratigraphic correlation (Rovere et al., 2014). Further details on the geology, geomorphology, and structure of this scarp are found in Rovere et al. (2014). Given the availability of SIS geochronology, we classify the age control of this scarp as

strong (Fig. 2). The De Hoop (South Africa) scarp is mapped with a combination of high-resolution remote sensing, TerraceM analysis, and direct field measurements. In total, the scarp extends ~59 km and varies from 12.4 m to 37.4 m in elevation (Fig. 5g). Lacking a clear, first-order pattern of short-wavelength variability, the toe of the De Hoop (South Africa) scarp occurs at ~28 m with consistent (but small) variability above and below this elevation across its full length. Due to the presence of colluvial deposition along the scarp's toe, our TerraceM analysis provides important downward correction on the order of ~10

m (Fig. 4g).

### 3.1.8 Socotra (Yemen)

The Socotra (Yemen) scarp is located on the island of Socotra ~360 km off the coast of mainland Yemen; the scarp is situated on the island's southern coast (Fig. 3h). The coastal plain consists of Plio-Quaternary sedimentary units that are interrupted by the limestone-dominated Eocene upland with a sharp cliff (Beydoun & Bichan, 1969; Pik et al., 2013). The Pliocene coastal

plain sediments consist of horizontal terraces with conglomerate beds of marine fossils overlying older bedrock (Beydoun & Bichan, 1969). Schlüter (2006) describes prominent alluvial, colluvial, talus, dune, and beach deposits along the southern coast of Socotra, bordered to the north by Cretaceous limestones and marls. Several rivers dissect the scarp along its ~75 km length (Fig. 3h). We characterize this scarp as having moderate age control due to the availability of geologic maps but the lack of absolute dates (Fig. 2). Although the island of Socotra had undergone the majority of its post-rift adjustment by Miocene times

following its separation from Arabia, this site represents one of the most dynamic tectonic settings included in this dataset, as it is situated proximal to the seismically active Arabia-India-Somalia triple junction (Birse et al., 1997; Fournier et al., 2001, 2010). The Socotra (Yemen) scarp, which is mapped in high resolution and analyzed with TerraceM (Fig. 4h), is characterized by a pattern of short-wavelength variability (~50 to 70 km, spatially). The maximum scarp elevation occurs at ~25 km distance at 74.9 m, after increasing from its lowest point (19.6 m) at start of the scarp trace to the west. The elevations descend gradually

from the maximum elevation to the end of the trace (east), where elevations return to ~26 m (Fig. 5h).

### 3.1.9 Nome (Alaska, USA)

The Nome (Alaska, USA) scarp is located on the southern coast of the Seward Peninsula in Alaska (Fig. 2). The town of Nome is situated on the coast facing the Bering Sea, ~4 km from the scarp. The coastal plain is covered with glacial and





undifferentiated deposits, the latter of which is likely sourced from unglaciated uplands and lowlands (Péwé, 1975). Glaciation
from MIS 6 likely extended to the Nome coastal plain; however, multiple sea-level highstands from late Pliocene to Pleistocene
times also left imprints on the coastal plain (Kaufman & Brigham-Grette, 1993). The high elevation terrain that composes the
scarp cliff rocks include Devonian to Ordovician schist and marble (Till et al., 2009). The oldest marine deposits mapped on
the coastal plain are linked to the Beringian transgression (2.7 to 2.5 Ma). We characterize this scarp with moderate age control
as the site has been well-mapped, and the relative geochronology provides a more robust constraint in comparison to other
sites (Fig. 2). Due to a lack of SRTM coverage above 60˚N, the Nome (Alaska, USA) scarp was mapped in lower resolution
and was not analyzed with TerraceM. Extending ~29 km in length, the Nome scarp varies in elevation from 50.0 m to 84.0 m.
The mapped elevations remain relatively constant (~50 m) along the length of the scarp, then increase at the eastern margin of
the scarp trace to 84.0 m (Fig. 5i).

### 3.1.10 East Coast (USA)

The East Coast (USA) scarp, also known as the Orangeburg Scarp, represents the longest and most studied scarp in this dataset
(Fig. 2; e.g., Dowsett & Cronin, 1990; Rovere et al., 2014, 2015; Winker & Howard, 1977). To the south, the scarp's coastal
plain is composed of the Duplin Formation, a warm, shallow marine and inner to middle-shelf facies. To the north, the plain
consists of the Raysor Formation, which corresponds to a slightly deeper marine depositional environment (Huddlestun, 1988).
Further details on the this scarp's geology, geomorphology, and structure can be found in Rovere et al. (2014) and Rovere et
al. (2015). Given the presence of SIS ages (2.30 Ma to 3.57 Ma; Graybill et al., 2009; McGregor et al., 2011) and well-
developed biostratigraphic constraints (e.g., Dowsett & Cronin, 1990), we classify this scarp's age confidence as strong (Fig.
2). The East Coast (USA) scarp is mapped in high resolution and with direct field measurements. A TerraceM analysis was
not performed for this site since it has been extensively mapped based on DEMs and field observations as described in Rovere
et al. (2015). Along its 1089 km length, this scarp exhibits long-wavelength patterns of variability with elevations ranging
between 23.0 m and 92.0 m. From south to north, the East Coast (USA) scarp increases in elevation until ~520 km distance to
~80 m, before decreasing modestly (by ~30 m) until ~660 km distance (Fig. 5j). The scarp then increases again to its maximum
point ~925 km distance and decreases again by ~25 m for another ~160 km distance to the northern terminus of the scarp (Fig.
5j).

### 3.2 GIA correction

The Nome (Alaska, USA) site is most significantly affected by GIA, with a mean (mean of the mean and mean of the standard
deviation across Earth models) of $6.39 \pm 0.11$ (1$\sigma$) m, as it is located on the actively subsiding peripheral bulge of the former
Laurentide Ice Sheet. The East Coast (USA) scarp is also affected by ongoing solid Earth adjustment from the collapsed
Laurentide Ice Sheet, with a mean correction of $3.33 \pm 4.69$ (1$\sigma$) m along the scarp trace. At this site, the correction transitions
from positive to negative moving north along the scarp, which is a result of peripheral bulge subsidence in the south and solid



Earth rebound in the north (Fig. 7a). These two sites will be most sensitive to the ice history chosen in the GIA correction, which is not varied in this study. All of the remaining sites are located in the far field of ice loading changes and have corresponding GIA corrections characterized by lower uncertainty (Fig. 7b). Continental levering, a mechanism that occurs due to loading and unloading of water on the continental shelf and the flexural response of the lithosphere along coastlines, is the primary driver of adjustment at the remaining sites. Mean corrections and 1σ uncertainties at the Kongo Central (DRC; -

$3.9 \pm 0.03$ m), De Hoop (South Africa; $-2.26 \pm 0.05$ m), Mahafaly (Madagascar; $-1.20 \pm 0.10$ m, Benghazi (Libya; $-1.17 \pm 0.53$ m), Al Wusta (Oman; $-3.92 \pm 0.08$ m), Socotra (Yemen; $-2.36 \pm 0.07$ m), Darling (Australia; $-0.22 \pm -0.61$ m), and Roe Plain (Australia; $-2.50 \pm 0.12$ m) sites also have minor contributions from perturbations to Earth's rotation axis (e.g., Australian sites) and equatorial syphoning (e.g., Kongo Central site), which occurs when local sea level falls in response to distal infilling of subsiding peripheral bulges. At all sites, the GIA correction does not fully account for the amplitude of observed elevations

or the spatial elevation variability of the mapped scarps (Figs. 7a & 8), and ultimately it is a minor contributor to scarp elevation in comparison to DT.



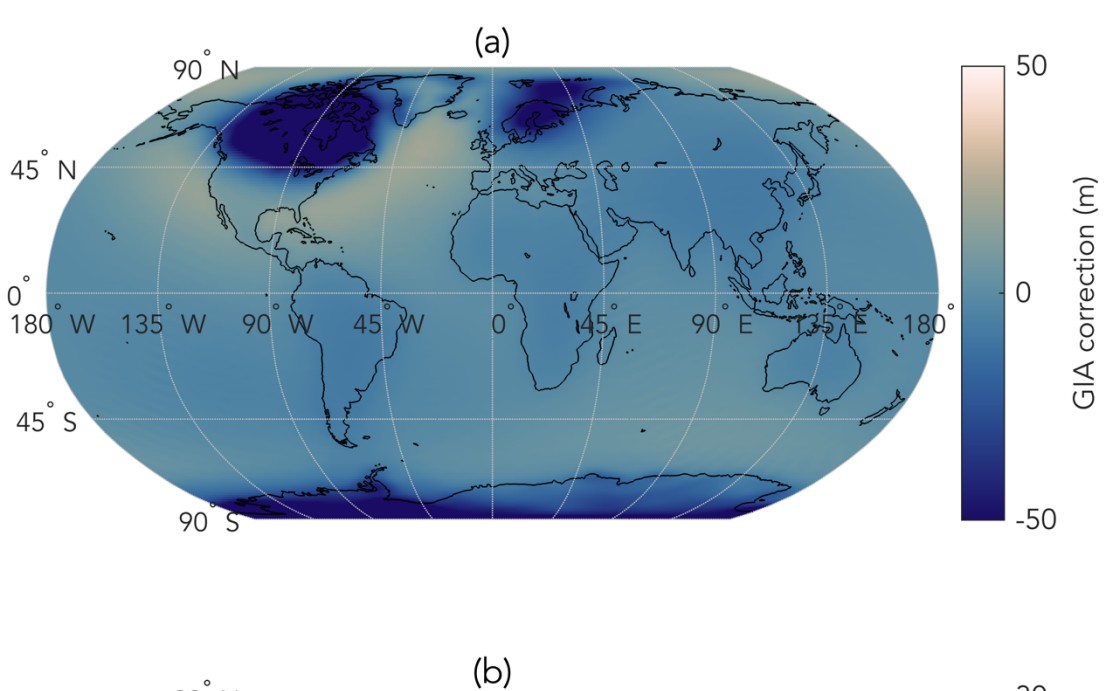

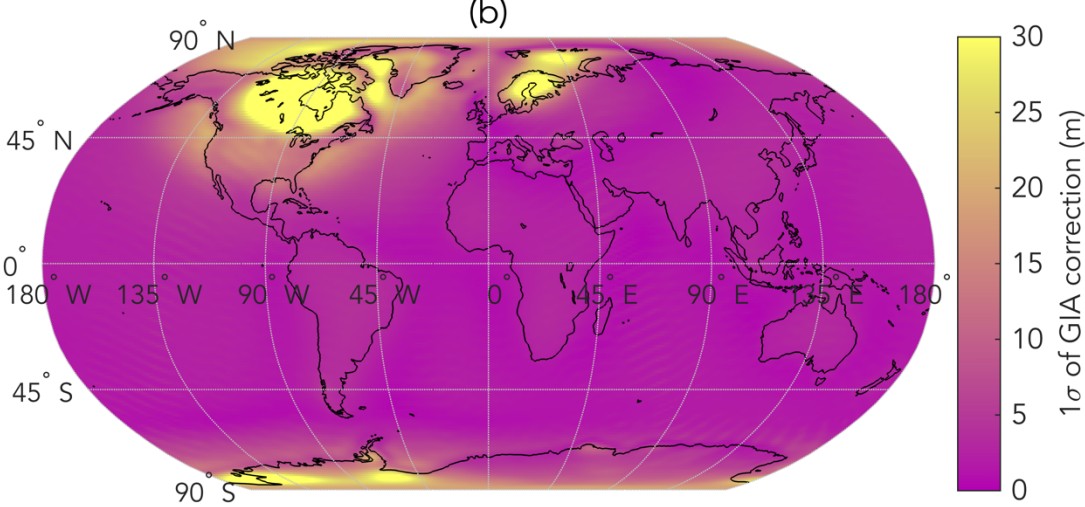

**Figure 7:** GIA correction and 1σ uncertainty associated with the assumed 1-D Earth structure based on Pleistocene-to-present disequilibrium computed by Raymo et al. (2011). (a): Mean GIA correction based on 36 radially symmetric viscosity structures. (b): One standard deviation (1σ) of the mean correction.

### 3.3 DT change correction

Our full suite of DT change predictions over the last 3 Myr yields highly variable spatial patterns of deformation across the globe with amplitudes of up to ~ ± 200 m (Supplementary Fig. 2). This variability is due to uncertainties in the thermal and viscosity structure of the mantle as well as uncertainties in plate motions. As a result, no individual model prediction simultaneously matches the observed residual deformation after correcting for GIA at all scarp locations (Fig. 8), precluding



a unified inference of Pliocene GMSL from any one DT model. At five of the ten sites, subsets of the DT change suite meet the stringent MSWD (<20) and GMSL (10 m to 40 m) criteria: Kongo Central (DRC), Mahafaly (Madagascar), De Hoop (South Africa), Socotra (Yemen), and Nome (Alaska, USA; Fig. 8c,e,g,h,i). Just one model meets the stringent criteria at the Kongo Central site and predicts southward upward tilting in agreement with the observations (Fig. 8c). While this model falls
within the range of data in the north, it begins to underpredict deformation in the south. At the Mahafaly (Madagascar) site, a greater subset of models (n = 8) meets the stringent criteria; all of these models predict uniform elevation change along the scarp within the range of the data (Fig. 8e). The subset of DT models that meets the stringent criteria at the De Hoop (South Africa) site (n = 3) predicts a subtle decrease (~5 m) in elevations from west to east, which is somewhat consistent with the data, which show mostly uniform elevations along the scarp's length (Fig. 8g). In Socotra (Yemen), one model fulfills the
stringent criteria but fails to capture the short wavelength variability that occurs at ~30 km distance (Fig. 8h). The largest subset (n = 15) of models that meets the stringent criteria occurs at the Nome (Alaska, USA) site, but no model in this subset captures the stark increase in elevation at the easternmost point of the scarp (Fig. 8i).

At the Darling (Australia), Roe Plain (Australia), Benghazi (Libya), Al Wusta (Oman), and East Coast (USA) sites, no model achieves our stringent MSWD and GMSL criteria. When applying the weak criteria (MSWD <30 and GMSL
between 0 m and 50 m), these sites coincide with small subsets of the DT change suite (excluding Benghazi, Libya, which does not fit even the weak criteria). At the Darling (Australia) site, the subset (n = 3) that fulfills the weak criteria falls within the range of the observations in the south but fails to capture the assumed uplift (~40 m) in the north (Fig. 8a). At the Roe Plain (Australia) site, the weak criteria subset (n = 4) predicts linear patterns of deformation but not the anticlinal structure apparent in the observations (Fig. 8b). At the Al Wusta (Oman) site, the weak criteria model subset (n = 4) predicts first-order agreement
with the data with slightly higher elevations (~10 m to 20 m) in the south that transition to lower elevations (~10 m) in the north (Fig. 8f). Finally, the East Coast (USA) scarp is consistent with a weak criteria model subset (n = 4) that predicts gradual uplift from ~0 m to ~40 m over ~600 km distance, which is consistent to first order with the spatial pattern of deformation but underpredicts the amplitude of change from the observations (Fig. 8j).

By applying this filtering criteria, we identify at each site which convection parameters are preferred (Fig. 8). A
systematic preference for a radial reference viscosity and plate rotation correction is lacking; however, scarp locations appear sensitive to the specific upper mantle structure that is assigned in the model. The Darling (Australia), Roe Plain (Australia), and the East Coast (USA) sites, where only the weak fitting criteria were met by DT change predictions, all employ the upper mantle structure from the SL2013sv tomography model (Fig. 8a,b,j; Schaeffer & Lebedev, 2013). At the Mahafaly (Madagascar), Al Wusta (Oman), and Socotra (Yemen) sites, where models meet both the weak and stringent criteria, the best-
fitting models are parameterized exclusively with upper mantle structure from the TX2008 model (Fig. 8e,f,h; Simmons et al., 2009). No site contains best-fitting models constrained exclusively by upper mantle structure from GLAD-M25; however, the best-fitting models for Kongo Central (DRC), De Hoop (South Africa), and Nome (Alaska, USA) consist of upper mantle structures from both GLAD-M25 and SL2013sv (Fig. 8c,g,i; Lei et al., 2020; Schaeffer & Lebedev, 2013).



**Figure 8: Comparison of GIA-corrected scarp deformation with predictions of DT change. Red points indicate mapped scarp elevations corrected for GIA. Their uncertainties are the square root of the sum of the squares of the individual uncertainties from the indicative range, field measurements, TerraceM analysis (Jara-Muñoz et al., 2016), and the GIA correction. Gray lines show Gaussian Process Regression models trained with the GIA-corrected elevation data. Dark and light gray bands correspond to 1σ and 2σ uncertainties of the Gaussian model, respectively. Green, blue, and red lines show predictions of DT change with upper mantle structures from the GLAD-M25, SL2013sv, and TX2008 tomography models, respectively (Lei et al., 2020; Schaeffer & Lebedev, 2013; Simmons et al., 2009). Solid colored lines correspond to DT change solutions that fit the stringent fitting criteria. Dashed colored lines correspond to DT change solutions that meet the weak fitting criteria. The inferred GMSL and fully propagated uncertainty is shown for sites that meet the stringent fitting criteria. (a) Darling (Australia); (b) Roe Plain (Australia); (c) Kongo Central (DRC); (d) Benghazi (Libya); (e) Mahafaly (Madagascar); (f) Al Wusta (Oman); (g) De Hoop (South Africa); (h) Socotra (Yemen);**



**(i) Nome (Alaska, USA); (j) East Coast (USA); (k) map inset showing site locations superimposed over the 1σ uncertainty from the full DT change model suite.**

## 4 Discussion

With the dual motivation of (1) improving the accuracy of mantle convection simulations and (2) using predictions of mantle DT change to correct for solid Earth deformation to infer Pliocene GMSL, we present a dataset of ten globally distributed scarps that record ancient sea-level highstands. Data-model comparison demonstrates that our current generation of convection models is incapable of accurately simulating deformation at each site simultaneously, and in the following sections we explore several reasons for data-model misfit and recommendations for future work.

### 4.1 Data-model misfit due to data uncertainties

While this dataset is a result of rigorous analysis of each scarp's geomorphology, in which we use high-resolution DEMs, direct field mapping, and secondary analysis of landscape evolution, a considerable amount of subjectivity can be involved in interpreting sometimes complex geologic archives of past sea level. For example, while many coast-perpendicular topography profiles are characterized by easily-identified platform and cliff geometries (e.g., the Al Wusta (Oman) and Roe Plain (Australia) sites), many others are not (e.g., the Mahafaly (Madagascar) site) and involve uncertain choices about defining scarp structure. One important cause of uncertainty is the presence of dunes on the coastal plain, which if not identified properly, can lead to an overestimation of the platform elevation. Field mapping reduces the subjectivity involved in analyzing DEMs; however, many of the scarps included here were not mapped through direct observations. In some cases (e.g., the Benghazi (Libya) site), a cliff is well defined in cross section, but the platform appears mostly buried by surface sediments, obfuscating its structure (Fig. 4d). At other locations, such as the Kongo Central (DRC) site, the platform is well defined but shows high-frequency variability in cross section that increases uncertainties in the extrapolated intersection (i.e., scarp toe; Fig. 4c).

In addition to challenges associated with characterizing the structure of the scarps, age uncertainties play a vital role in the comparison between scarp elevation data and geodynamic predictions. All of the convection simulations that we present correspond to DT change over the last 3 Myr, which approximately corresponds to deformation since the MPWP. That said, the true formation age of the scarps may be challenging to discern without an absolute geochronology. As predicted by the Trenhaile (2014) model framework, high frequency, orbitally paced sea-level oscillations would have carved away steep cliff faces and deposited horizontal coastal plains. However, these sea-level highstands may not have affected every site equally, owing to geographically variable sea-level change and geodynamic conditions. To preserve the sea-level marker, this formational model depends on a background geodynamic state where the crust undergoes either no uplift or slow rates of uplift; subsiding regions may cause erosion platforms to be submerged and ultimately not preserved in the topography. At sites where data-model misfit is great (e.g., the Benghazi (Libya) site), the age uncertainty may in part explain the misfit; however, the misfit does not provide decisive information about whether the true age of the scarp is younger or older than the simulation.



On the other hand, a low data-model misfit may indicate a scarp's formation age is indeed close to 3 Ma (e.g., the De Hoop (South Africa), Kongo Central (DRC), and Al Wusta (Oman) sites).

**4.2 Data-model misfit due to model uncertainties and proposed improvements**

After correcting each scarp for GIA, a process that alone cannot explain all of the observed deformation, we compare the residual deformation with predictions of DT change. Despite the success of some predictions of DT change, nearly all sites within this dataset are characterized by shorter wavelength elevation variability than is simulated by our models (Fig. 8). This is most apparent at the Socotra (Yemen) scarp, where the scarp's elevations reflect an anticlinal geometry over <80 km distance (Fig. 8h) that is not seen in the DT model predictions. Slightly longer wavelength (150 km to 500 km) anticlinal structures are also apparent at the Darling (Australia), Roe Plain (Australia), and East Coast (USA) sites, and our preferred models at those locations yield only moderate success at reproducing this short-wavelength variability (Fig. 8a,b,j). Importantly, the wavelength of DT is directly related to the wavelength of the particular upper mantle tomography model used in our convection simulations; SL2013sv consists of the shortest wavelength mantle structure that is included in our convection suite and performs well for most sites in the dataset.

By generating a large suite of convection simulations, we are able to assess which localities favor certain model parameters. We find that the upper mantle buoyancy structure has a first-order impact on the success of our models at any given location (Fig. 3). For example, only models constrained by the TX2008 tomography model produce realistic predictions of along-scarp deformation and inferred GMSL at the Mahafaly (Madagascar), Al Wusta (Oman), and Socotra (Yemen) scarps (Fig. 8e,f,h). These locations are all in the proximity of the African superplume (Ni et al., 2002; Simmons et al., 2007). The TX2008 model is the only one in our suite in which density perturbations were tuned to fit present-day surface observations (e.g., the global free-air gravity field, tectonic plate divergences, and DT). This model employs laterally varying correction factors to the shear wave velocity-to-density scaling, a procedure that constrains important compositional variation between mantle plume structures (as well as cratonic roots) and the ambient mantle. Our simulations constrained by SL2013sv and GLAD-M25-derived upper mantle structure are not well suited for these sites, as our conversion from seismic velocity-to-temperature neglects the compositional difference of superplumes from the surrounding mantle; this leads to upwelling above plume structures in these models that are too fast. That said, many other sites, including the Darling (Australia), Roe Plain (Australia), Kongo Central (DRC), and East Coast (USA) scarps are only compatible with models parameterized with SL2013sv-derived upper mantle structure (Fig. 8a,b,c,j). This model is the only one that also uses surface waves in the inversion, which resolves shorter wavelength upper mantle structure that appears to be important to predict DT change along these passive margins.

Our data-model comparison provides insight into how models can be improved. As the present-day 3-D buoyancy structure has a first-order influence over the convective regime, improving this constraint by incorporating regional tomography models or those that employ surface waves will be critical. The calibrated parameterization of the Yamauchi &



Takei (2016) shear wave velocity-to-temperature conversion by Richards et al. (2020) provides a necessary constraint on the effects of anelasticity on seismic velocity and ultimately on the inferred properties of the mantle. However, improvements to this conversion that invoke new data constraints can serve to better predict mantle temperatures, densities, and viscosities. In addition, accurate parameterization of compositional structures (e.g., lithosphere and Large Low-Shear-Velocity Provinces) will have important effects on DT predictions (Richards et al., 2023). While our models incorporate lateral viscosity variations,

they likely underestimate the true amplitude of variability (Yang & Gurnis, 2016), and while increasing this contrast comes with computational expense, it remains a worthwhile objective. Compressibility, phase transitions, and non-linear rheology are some further improvements that invoke more complex physics and will improve the fidelity of DT results as well (Colli et al., 2018; Tackley, 2008).

### 4.3 Data-model misfit due to potential auxiliary deformation processes

In addition to data and model deficiencies, auxiliary processes, which include flexure and isostatic adjustment due to sedimentary load changes (e.g., Moucha & Ruetenik, 2017) or brittle deformation in the crust (i.e., faulting), are potential sources of short-wavelength variability in the topography. These auxiliary processes are not explicitly accounted for in our modeling framework. For example, the Socotra (Yemen) and Benghazi (Libya) scarps occur peripheral to active tectonic boundaries, and while compelling evidence of crustal deformation would require better field observations, these sites have

likely been deformed at least in part by tectonic processes. This factor may also be related to their higher mapped elevations in comparison to other sites in the dataset (Fig. 5 & 8). While the Darling (Australia) site also falls within this higher elevation range (maximum elevation >100 m) and is situated near-parallel to the prominent Darling Fault (Fig. 3a), it is not likely that crustal processes deformed the scarp since the Darling Fault was last active during times much earlier than is relevant to the formation of the scarp (Fletcher et al., 1985). Other sites, such as the East Coast (USA), exhibit considerable residual

deformation after both the GIA and DT change corrections have been applied. This is likely a result of sedimentary loading and crustal flexure, which are not accounted for in our modeling framework but are thought to be important processes for this region (Moucha & Ruetenik, 2017). The Al Wusta (Oman) site is located peripheral to salt diapirism, a regional process that occurs on million year-timescales and may be relevant to the site's recent deformation (Li et al., 2012). Diffusive landscape evolution at most sites also contributes to the observed topography, but we attempt to correct for this through our TerraceM

analysis (Fig. 4). All of these effects occur synchronously with DT change and must be considered to unravel the full deformational history of any site, emphasizing the need to consider regional processes.

### 4.4 GMSL inference and implications

While data and model uncertainties persist, there are five scarps for which along-scarp deformation and the magnitude of deformation in the models are consistent (Fig. 3; solid lines): Kongo Central (DRC), Mahafaly (Madagascar), De Hoop (South

Africa), Socotra (Yemen), and Nome (Alaska, USA). We calculate the inferred GMSL for each site using the DT prediction



that yields the best fit (lowest MSWD) to the along-scarp deformation. These locations show minimal residual deformation after all corrections have been applied, suggesting that auxiliary processes may not have played a major role in their deformation since 3 Ma. These five sites produce GMSL estimates of 13.1 ± 10.1 m (Kongo Central, DRC), 10.1 ± 8.2 m (Mahafaly, Madagascar), 11.6 ± 5.2 m (De Hoop, South Africa), 16.9 ± 14.7 m (Socotra, Yemen), and 10.8 ± 18.6 m (Nome,

Alaska, USA; Fig 8c,e,g,h,i). The De Hoop (South Africa) site in particular has comparatively good age control (3.56 ± 1.08 Ma; Rovere et al., 2014) and the lowest GMSL uncertainty, which makes this our most reliable constraint. The preferred model also predicts patterns of present-day DT consistent with observations (present-day DT varying from ~100 m to >500 m; Hoggard et al., 2017). The remaining four sites are characterized by greater GMSL uncertainty and misfit with residual topography observations.

The De Hoop (South Africa) mean estimate for MPWP GMSL is lower than the inference from Mallorca (16.2 m; 3.27 ± 0.12 Ma) but falls within the reported 1σ uncertainty (5.6 m to 19.2 m; Dumitru et al., 2019). If valid, this lower range may suggest ice sheets were relatively stable even under warm Pliocene climate conditions (2.5˚C to 4˚C above 1850 to 1900 baseline; Fedorov et al., 2013; Fischer et al., 2018; Haywood et al., 2013). With the GIS contributing ~7 m (Bierman et al., 2016; Morlighem et al., 2017), WAIS contributing ~3.2 m from its most unstable sectors (Bamber et al., 2009), and thermal

expansion contributing ~1.5 m (Dumitru et al., 2019), little to no excess melt contribution from the EAIS is required to fit this Pliocene sea-level budget.

**5 Conclusion**

We present a dataset of ten wavecut scarps that formed during Pliocene times, when Earth's mean temperatures were similar to and higher than present-day levels. This interval is a critical target for sea-level reconstructions because it can be used to

calibrate projections of sea-level rise this century (e.g., DeConto et al., 2021). We use a combination of high-resolution remote sensing, direct field mapping, and landscape evolution analysis (i.e., TerraceM) to characterize the topography of each scarp. These globally distributed sites show patterns of short-wavelength variability in the topography, which has resulted from ~3 Myr of solid Earth deformation primarily due to GIA and DT change as well as brittle deformation in the crust, flexure due to sedimentary load changes, and surface deposition or incision. As deformation due to GIA is characterized by significantly

lower amplitude and uncertainty, especially in the far field of Pleistocene ice sheets, DT change represents a much more uncertain correction owing to the many under-constrained parameters involved in simulating mantle convection. We compare a large suite of DT change predictions (n = 135) to the scarp dataset and find that no individual DT change model can accurately predict epeirogenic motion at every site; however, model fits exist for selected sites. This data-model comparison informs proposed model improvements, largely focused on better constraining upper mantle structure and rheology. Despite persistent

uncertainties in both the data and models, we use our best-fitting predictions to remove the effects due to GIA and DT change and compute GMSL estimates. Our preferred GMSL inference (11.6 ± 5.2 m) from the De Hoop (South Africa) site falls in the lower range of existing estimates for the MPWP. If valid, this would suggest ice sheets may have been more resistant to



the warm Pliocene climate conditions than previously thought. Nonetheless, Pliocene results confirm that Earth's present-day ice sheets are increasingly out of equilibrium with heat and greenhouse gas in the atmosphere, likely leading to multi-meter sea-level rise in the coming centuries.

**Acknowledgements**

We recognize Jonathan Gale, Mike O'Leary, and Alex Janßen, whom helped locate and remap the seven additional scarp locations. The authors acknowledge NSF grant OCE-1202632 'PLIOMAX' for support. JA acknowledges funding from the Alfred P. Sloan Research Fellowship FG-2021-15970. FR thanks the Imperial College Research Fellowship and Schmidt Science Fellowship schemes. MH acknowledges support from the Australian Research Council DECRA DE220101519 and the Australian Government's *Exploring for the Future* program. AR acknowledges support from the European Research Council (ERC) under the European Union's Horizon 2020 research and innovation programme (grant agreement n. 802414). The authors acknowledge PALSEA, a working group of the International Union for Quaternary Sciences (INQUA) and Past Global Changes (PAGES), which in turn received support from the Swiss Academy of Sciences and the Chinese Academy of Sciences. We acknowledge computing resources from Columbia University's Shared Research Computing Facility project, which is supported by NIH Research Facility Improvement Grant 1G20RR03893-01, and associated funds from the New York State Empire State Development, Division of Science Technology and Innovation (NYSTAR) Contract C090171, both awarded 15 April 2010. We thank the Computational Infrastructure for Geodynamics (geodynamics.org) which is funded by the National Science Foundation under award EAR-0949446 and EAR-1550901 for supporting the development of ASPECT.

**Author contribution**

MR, AR, AH, and JA conceptualized the project. MR, JA, and AR were responsible for funding acquisition. AH, JA, and AR performed the formal analysis. FR and MH provided assistance with convection model development. All authors contributed to investigation as well as writing and editing the manuscript.

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
