# Peer review of "Pliocene shorelines and the epeirogenic motion of continental margins: A target dataset for dynamic topography models"

_EGUsphere, 2023_

## Author Comment (AC1)

Dear Editor and Referees,

We thank you for your close review and constructive feedback on our study. In this response, we provide detailed answers to the questions raised. Below, we have interspersed each Referee comment in black and our responses in blue. Line numbers throughout this response refer to the original submission.

Best,

The Authors

**Referee #1:**

1. Pliocene shorelines and the epeirogenic motion of continental margins: A target dataset for dynamic topography models by Hollyday et al. is a well written, fully referenced, attempt to address a most difficult research objective in a comprehensive manner. Identifying the response of Pliocene shorelines is a challenging activity that the research team pursues efficaciously. The results are informative and lead to further study that may reveal additional factors affecting topography.

As they note, "We next determine the signal associated with mantle dynamic topography by back-advecting the present-day three-dimensional buoyancy structure of the mantle and calculating the difference in radial surface stresses over the last 3 Myr using the convection code ASPECT. We include a wide range of present-day mantle structures (buoyancy and viscosity) constrained by seismic tomography models, geodynamic observations, and rock mechanics laboratory experiments." Back-advecting the present-day three-dimensional buoyancy structure of the mantle and calculating the difference in radial surface stresses over the last 3 Myr involves the application of complex mathematical models in an effort to approximate mantle buoyancy and viscosity that play a role in topographic evolution. Hollyday et al. are to be complimented for recognizing the possible role dynamic topography and, furthermore, for the mathematical prowess that enables an attempt to constrain the driver(s) of dynamic topography. The paper documents a valiant effort!
Citation: https://doi.org/10.5194/egusphere-2023-2099-RC1

We thank Referee #1 for their review of the manuscript. They note how the study leverages a range of constraints on mantle structure in combination with state-of-the-art numerical models for simulating mantle convection to make predictions of dynamic topography (DT) change over millions of years. As the Referee points out, by employing 3-D mantle convection simulations and comparing model predictions with a global dataset, we can probe the possible role mantle dynamics has on topographic evolution over millions of years.

**Referee #2:**

1. This manuscript presents a global dataset of ten wavecut scarps formed by successive Pliocene sea-level oscillations that are presently between 6 and 109 m above sea level. Correcting for glacial isostatic adjustment does not explain the observed difference in Pliocene paleoshoreline elevations around the globe. The main aim of the manuscript is to investigate whether mantle-flow-driven topography can explain these differences. A series of 27 backward advected mantle flow models is presented across which the radial viscosity and 3-D buoyancy structure are varied. The 3-D buoyancy structures were obtained from two tomographic models in the upper mantle and four tomographic models in the lower mantle. A plate correction is applied to the dynamic topography models, resulting in 135 models in total. Two criteria (either weak or stringent) are used to assess the success of dynamic topography models: the Mean Weighted Standard Deviation (MSWD) of the GIA- and DT-corrected scarp elevations, and the inferred GSML value for the dynamic topography model. The results suggest that while dynamic topography likely contributes to sea level change over these time scales, global mantle flow models do not predict dynamic topography changes at the scale of the inferred topographic oscillations along scarps (tens to hundreds of kilometres). An important contribution of the manuscript is to expose and discuss the limitations of the mantle flow models. My main recommendation is to provide more details and representations of the dynamic topography models to make the contribution accessible to readers who are not experts on this topic.

The results are promising in terms of deriving Pliocene paleoshoreline elevations around the globe by combining global digital elevation models with geological observations and data. The presented glacial isostatic adjustment correction includes error propagation and considers 36 radially symmetric viscosity structures. The predictions of the considered dynamic topography models significantly differ from one another, and between 1/135 and 15/135 models satisfy the stringent criteria at five of the ten considered sites. At four sites, between 3/135 and 4/135 models satisfy the weak criteria. There is one site for which none of the considered models succeeds.

I found the discussion to be appropriate and balanced. It mentions that other tectonic processes (flexure, deformation) may be at play and that an important limitation is that tomographic models are too smooth to resolve the topographic trends observed along GIA-corrected scarps. Despite the limited success of the dynamic topography models, it is noted that there are five scarps for which the mantle flow models predict consistent GMSL estimates. Results are emphasised for the well-dated De Hoop scarp (for which 3/135 dynamic topography models succeeded), which leads to a GIA- and DT-corrected GMSL 11.6 ± 5.2 m, a range that would require little melting of ice sheets under warm Pliocene climate conditions.

I appreciate that the limitations and uncertainties of the mantle flow models are exposed. Dynamic topography predictions are essential to the manuscript, however, no global maps of such predictions are shown. It would be helpful to show maps of dynamic topography and/or of the change in dynamic topography either globally and/or at some or all of the ten sites. It would also be helpful to represent the considered parameter space for the flow models graphically, and which models succeed or fail within that parameter space. This could be done across a series of X-Y plots or using some more sophisticated visualisation.

We would like to thank Referee #2 for their thorough and constructive review of our study. We agree with the Referee's summary of the objectives, methodologies, and key results of the study. One minor correction to their comment is that our convection suite includes a whole mantle model from TX2008 (Simmons et al., 2009) in addition to composite models based on two upper mantle and four lower mantle tomographic constraints. We also acknowledge the Referee's recommendation to provide more representations of DT model predictions. In the revised manuscript, we now include two additional figures to further visualize our results: (1) Four global maps showing the mean of the predicted present-day DT across our convection suite and its 1σ uncertainty as well as the mean of the change in DT over 3 Myrs and its 1σ uncertainty (this will be located in the main text), and (2) a series of five regional maps for scarps where we report a Pliocene GMSL inference showing our best-fitting models of DT change over 3 Myrs (this will be located in the supplementary materials). We also appreciate the Referee's suggestion of adding a visualization of the convection parameters space. In the revised manuscript (located in the main text) we now include a figure showing heatmaps of the mean MSWD (across the five total rotations) for each scarp location for every mantle temperature and viscosity pair. In this figure, green circles overlay parameter combinations where at least one rotation constraint leads to a GMSL inference between 10 m and 40 m.

[Figure]

Figure 8: Present-day and change in DT over 3 Myrs model predictions. (a) Mean of present-day DT across convection suite (n = 27); (b) 1σ of the mean present-day DT prediction; (c) the mean of the change in DT over 3 Myrs across the convection suite (n = 135); (d) 1σ of the mean DT change over 3 Myrs with plate motion corrections applied.

[Figure]

Supplementary Figure 3: Maps of best-fitting DT change for scarp locations that meet the stringent criteria and for which GMSL is reported. Red lines indicate scarp traces. (a) Kongo Central (DRC); (b) Mahafaly (Madagascar); (c) De Hoop (South Africa); (d) Socotra (Yemen); (e) Nome (Alaska, USA).

[Figure]

Figure 10: Heatmaps of convection suite parameter space and mean MSWD across the five total rotations. Green circles indicate viscosity-Earth model pairs where at least one total rotation correction leads to a GMSL inference that falls within the range of 10 m to 40 m. (a) Roe Plain (Australia); (b) Darling (Australia); (c) Kongo Central (DRC); (d) Benghazi (Libya); (e) Mahafaly (Madagascar); (f) Al Wusta (Oman); (g) De Hoop (South Africa); (h) Socotra (Yemen); (i) Nome (Alaska, USA); (j) East Coast (USA).

2. While the sensitivity of the mantle flow models to viscosity and buoyancy is exposed, it would be good to remind the reader that the viscosity space is large. A figure showing the different viscosity structures considered in the study would be helpful, ideally with some context of previous work (e.g. Mao and Zhong, 2021, https://doi. org/10.1029/2020JB021561). Showing differences in the buoyancy structure would require some visualisations of the mantle flow models, perhaps at present-day. It would be helpful to mention that the buoyancy structure depends on the conversion factor from relative seismic velocities to relative density variations. While a reasonable choice was made for the study, other choices would be possible. It might also be possible to mention the adjoint approach for mantle flow modelling, and its possible suitability to the problem at hand.

We agree with the Referee that the manuscript would be improved by visualizing the range of viscosity constraints used in our model suite as well as a comparison with previous work. In the revised manuscript, we include a new figure (located in the supplementary materials) showing the three radial viscosity constraints used in our convection suite in comparison with the range of mantle viscosities reported by Mao & Zhong (2021). This figure illustrates that our viscosity range mostly falls within the range of inferred viscosities from the best-fitting geoid predictions of Mao & Zhong (2021).

[Figure]

Supplementary Figure 4: Radially symmetric reference viscosity profiles used in the convection suite overlain with mantle viscosity estimates from Mao & Zhong (2021). The reference viscosities are scaled laterally according to each respective tomography-derived mantle temperature model.

The Referee accurately mentions that, while this study explores some of the possible range, we do not exhaustively characterize the full uncertainty associated with the mantle's buoyancy structure. By including global maps of the means and 1σ DT predictions for the present day and change over 3 Myrs, we aim to better convey the range of mantle flow results and corresponding DT model uncertainty associated with modeling different mantle buoyancy structures. We do not plot the mantle buoyancy structures; however, we have made the mantle temperature models available in a public repository (see Data availability section). In the revised manuscript, we have also added at line 221: "We note that the choice of conversion factor directly influences the computed mantle buoyancy structure and while alternative conversions may be suitable, all lower mantle constraints are based on the same radially symmetric factor (except for the TX2008 model which is based on a joint seismic and geodynamic inversion)."

The revised manuscript also mentions at line 553 the potential suitability of the adjoint mantle convection equations for predicting flow and better understanding Pliocene scarp deformation: "Lastly, we highlight that geodynamic inverse frameworks, in particular the adjoint approach (e.g., Ghelichkhan et al., 2021), may be uniquely positioned to invert for viscosity and density structures in the mantle that produce the observed topographic change."

3. It is not obvious from the manuscript how GMSL (global mean sea level?) corrections are obtained from each mantle flow model. It seems that this is done at each site, which I find confusing because at one location one can infer relative sea level, not global sea level. I do not think it is sufficient to refer the reader to Hollyday et al. (2023) for this calculation.

Global mean sea level (GMSL) is computed at each site by first correcting the scarp elevations for glacial isostatic adjustment (GIA) and then DT change over 3 Myrs. Without these two corrections, any given site would represent relative sea level; however, by applying both corrections we account for global variability in sea level owing to the gravitational, rotational, and solid Earth response to changes in ice and ocean loads through time as well as change in topography and the geoid from convection in the mantle. The resulting difference between GIA- and DT-corrected scarp elevations represents a hypothetical GMSL offset. GMSL is computed for every DT change prediction. The mean squared weighted deviation (MSWD) is also computed for each DT change prediction on the basis of the GIA-corrected scarp elevations at each site. A high MSWD for any given DT change model and GIA-corrected scarp indicates the DT change prediction does a poor job at predicting the remaining deformation (after the GIA correction but before the DT change correction has been applied) and consequently leads to a GMSL inference with lower fidelity. We agree with the Referee and have added the GMSL calculation to the revised manuscript at line 240:

"The GMSL computed at any given point along a scarp is given by,

$$GMSL_{i,m} = \overline{GE_i} - DT_{i,m} \tag{1}$$

where the GIA-corrected elevation, $\overline{GE_i}$, is the observed elevation minus the GIA correction and $DT_{i,m}$ is the DT change prediction over 3 Myrs at the $i^{th}$ location along the scarp for model suite member, $m$. We compute single GMSL values at each scarp for every DT change model prediction as a weighted average, where the along-scarp weights, $w_i$, and resulting model-specific GMSL, $GMSL_m$, are given by,

$$w_i = 1 \Big/ \sigma^2_{GE_i} \tag{2}$$

$$GMSL_m = \frac{\sum_{i=1}^N (w_i \cdot GMSL_{i,m})}{\sum_{i=1}^N w_i}, \tag{3}$$

where $\sigma_{GE_i}$ is the square root of the squared sums of the GIA, elevation measurement, and indicative range uncertainty at each location, $i$, along a given scarp, and N is the total number of elevation measurements along a given scarp. We report the GMSL uncertainty as:

$$\sigma_{GMSL_m} = \sqrt{\frac{\sum_{i=1}^N w_i (GMSL_{i,m} - GMSL_m)^2}{\sum_{i=1}^N w_i}}, \tag{4}$$

where the weights are from Equation 2. Another way to quantify the success of a particular DT change model at reproducing along-scarp deformation is to calculate the mean squared weighted deviation (MSWD) of the GIA-corrected scarp elevations: the smaller the MSWD, the smaller the variability of GIA- and DT-corrected elevations and the higher the confidence in a specific DT model. The MSWD is given by,

$$\text{MSWD}_m = \frac{1}{N} \sum_{i=1}^{N} \left[ \frac{(\text{GMSL}_{i,m} - \text{GMSL}_m)^2}{\sigma_{\text{GE}_i}^2} \right].$$ (5)

"

4. I do not understand what plate motion correction is applied, and how it results in five new variants of each dynamic topography model (from 27 models to 135 models, L. 231-232). I do not think it is sufficient to refer the reader to Hollyday et al. (2023) for this correction.

We have clarified the manuscript at line 231: "Our DT change predictions apply a no-net-rotation correction based on Argus et al. (2011) as well as a total lithospheric rotation based on Zheng et al. (2014). We explore total lithospheric rotation uncertainty by applying five different rotation values based on the mean value (0.25° Ma⁻¹), 1σ (0.195 and 0.305° Ma⁻¹), and 2σ (0.14 and 0.36° Ma⁻¹). This postprocessing procedure increases the total number of DT change predictions to 135." Further details on this procedure can be found in Supplementary Figure 1 README.

5. How is the lithosphere defined in the models? It would be helpful to be explicit even though the potential role of the lithosphere is acknowledged in the discussion (see also Davies et al., 2019, https://doi.org/10.1038/s41561-019-0441-4). Could short-wavelength lithospheric thickness variations and associated flow explain some of the topographic observations along fault scarps?

Our conversions from seismic velocity to density and temperature within the lithosphere become inaccurate as they neglect the compositional variability between the mantle and lithosphere leading to unrealistically dense and cold predictions in the lithosphere. Following Jordan (1978), we set temperatures in the lithosphere equal to the depth average outside of the lithosphere, making it neutrally buoyant. For this we use tomography-specific maps of the lithosphere-asthenosphere boundary. Our models also set the lithosphere to a fixed viscosity of $1 \times 10^{22}$ Pa s. We have revised the manuscript to describe this procedure more explicitly at line 225: "Following Jordan (1978), we assume the lithosphere is neutrally buoyant by correcting temperatures within the lithosphere to the depth average outside of the lithosphere. We perform this correction using tomography-specific maps of the lithosphere-asthenosphere boundary across the full model suite, except for the TX2008 model as cratonic structure was already accounted for within that joint inversion (Simmons et al., 2009). Our models also set a fixed lithospheric viscosity of $1 \times 10^{22}$ Pa s (see Hollyday et al. (2023) for further details)."

Davies et al. (2019) provides key insights into the dual roles that lithospheric structure and mantle flow play in reconciling observations of oceanic residual topography measurements. Although the continental lithosphere is assumed to be neutrally buoyant in our simulations and

the ~200 km horizontal resolution of seismic tomographic models places a lower bound on the wavelength of lithospheric thickness variations we can simulate, our models do capture the influence of relatively short-wavelength changes in lithosphere-asthenosphere boundary topography on asthenospheric flow directions. A substantial fraction of the small-scale variations in our dynamic topography predictions will reflect this lithosphere-asthenosphere interaction but we suspect that some of our inability to fully reconstruct along-scarp elevation changes reflects the tomographic resolution limitations; although, other short-wavelength signals may also be responsible, such as sedimentary processes, flexure, and tectonic deformation. We mention this in the revised manuscript at line 553: "Our models capture the ability of short-wavelength lithospheric structure to promote small-scale convection and DT variations but are limited by the spatial resolution of tomography models."

6. L.582-583: 'The preferred model also predicts patterns of present-day DT consistent with observations (present-day DT varying from ~100 m to >500 m; Hoggard et al., 2017).' Should this criterion (fit to residual topography) be used to filter the predictions of all dynamic topography models? How many scarps is the preferred model consistent with? Which model is consistent with the most scarps?

While we make some first-order comparisons with residual topography, our models do not specifically target present-day DT observations. Instead, we aim to identify models that predict patterns of DT change since Pliocene times that are consistent with well-constrained along-scarp deformation. Global geodynamic observations (e.g., Earth's geoid and residual topography) provide critical means for both forward and inverse constraints on mantle structure, rheology, and dynamics; however, a full-scale integration of these datasets into our data-model comparison is beyond the scope of this study, which was undertaken to specifically target Pliocene scarp evolution. One important finding of this study is that no individual DT change prediction resolves scarp deformation across sites. This implies that our current generation of convection models must be regionally parameterized to reproduce the observed scarp deformation patterns. That said, it remains a worthwhile objective to identify which convection parameters are most consistent with observations in a global sense. In the revised manuscript, we include heatmaps showing the convection parameter space and mean MSWD at each site, with green circles overlaying predictions that lead to GMSL inferences within the likely range of 10 m to 40 m. This visualization emphasizes that parameter success is regionally specific. That said, many sites, particularly those far from known compositional heterogeneity or dynamic mantle structures (e.g., African superplume), prefer the SL2013sv upper mantle tomography model. Sites located near these structures are generally more consistent with model results constrained by the TX2008 model, which specifically accounts for compositional heterogeneity. We do not present a preferred model across all the sites; however, the DT change prediction parameterized with the TX2008 full mantle structure and the S40 radial reference viscosity with a total rotation of 0.36 $^{\circ}$ Myr$^{-1}$ leads to the lowest mean MSWD across the ten sites. We do not report this in the manuscript as our parameter analysis (e.g., heatmaps in main text) indicates parameter success, especially the choice of temperature model, is very regionally sensitive. That is, one individual simulation cannot appropriately predict deformation on a global scale, but a variable set of global parametrizations can fit many observations of Pliocene scarp topography.

7. Early in the manuscript, it seemed that scarps would only be constrained using remote sensing data and TerraceM. It became clear later on that remote sensing analysis was carried out for scarps for which geological and/or field observations are available. It would be helpful to mention this early on.

We use high-resolution digital elevation models (DEMs) from the Shuttle Radar Topography Mission (SRTM) and perform the TerraceM analysis for all sites except Nome (Alaska, USA) and East Coast (USA). At the Nome (Alaska, USA) site, we use a DEM from the General Bathymetric Chart of the Oceans (GEBCO) 2023 grid, as this site is outside of the range of SRTM. The East Coast (USA) has been extensively studied through direct field observations and high-resolution regional DEMs, so reported elevations are from Rovere et al. (2015), which we mention in the lines 402 – 403. In sections 3.1.1 – 3.1.10, we note whether field observations were made for each respective scarp. Supplementary Figure 1 also includes columns for the 'Data source' and 'Profile extraction method,' which specifies exactly how each elevation measurement was made. This is mentioned in lines 248 – 249. We have also added to line 83: "We characterize the topography of each site using remotely sensed and, in some locations, direct field observations."

8. L. 574: should this be Fig. 8?

Yes, we thank the Referee for catching this mistake. It has been corrected in the revised manuscript.

9. L. 588-591: what are GIS, WAIS and EAIS?

These abbreviations were not appropriately defined. We have removed them and provided their full form in the revised manuscript: Greenland Ice Sheet, West Antarctic Ice Sheet, and East Antarctic Ice Sheet.

10. Is GMSL defined in the manuscript?

We have now properly defined GMSL in line 32 in the revised manuscript.

11. There should be a space between consecutive units (see rates of change units)
Citation: https://doi.org/10.5194/egusphere-2023-2099-RC2

Thank you. We have now properly added spaces between consecutive units in the revised manuscript.